

# Ophthalmic nurses' knowledge, attitude, and practice toward venous thromboembolic prevention: a dual-center cross-sectional survey

Xiaoxi Zhou[1,*], Minhui Dai[1,2,3,*], Lingyu Sun[2,3], Chunyan Li[2,3], Wendi Xiang[4], Yaoyao Lin[5] and Dandan Jiang[5]

[1] Teaching and Research Section of Clinical Nursing, Xiangya Hospital of Central South University, Changsha, China
[2] Hunan Key Laboratory of Ophthalmology, Changsha, China
[3] Eye Center, Xiangya Hospital of Central South University, Changsha, China
[4] Department of Operating Room, Xiangya Hospital of Central South University, Changsha, China
[5] The Eye Hospital, School of Ophthalmology and Optometry, Wenzhou Medical University, Wenzhou, China
[*] These authors contributed equally to this work.

Corresponding author
Chunyan Li, 307909991@qq.com, lcy0203@csu.edu.cn

## ABSTRACT

**Background.** Venous thromboembolism (VTE) is a severe preventable complication among ophthalmic surgical patients. The knowledge, attitude, and practice (KAP) of nurses play a key role in effective VTE prevention. However, little is known about the KAP of ophthalmic nurses' VTE prevention. This study aimed to examine the level of KAP toward VTE prevention among Chinese ophthalmic nurses and to investigate the influencing factors of their VTE practice.

**Methods.** A total of 610 ophthalmic nurses from 17 cities in Hunan and Zhejiang Provinces, China, participated in this study. Data was collected *via* the Sojump online platform from March to April 2021. A self-administered VTE questionnaire was developed to assess nurses' KAP toward VTE prevention. Multiple linear regression analysis was used to analyze the influencing factors of ophthalmic nurses' VTE prevention practice.

**Results.** The scores (correct rates) of ophthalmic nurses' knowledge, attitude, and practice were $103.87 \pm 20.50$ (76.4%), $21.96 \pm 2.72$, and $48.96 \pm 11.23$ (81.6%), respectively. The three lowest-scored knowledge items were related to VTE complications, physical prevention, and risk assessment. The three lowest-scored attitude items were related to nurses' training, VTE risk, and patient education. The three lowest-scored practice items were related to the assessment scale, VTE assessment, and patient education. Nurses' knowledge, attitude, and practice were significantly correlated with each other. Multiple linear regression analysis showed that Hunan Province ($B = 2.77$, $p = 0.006$), general hospital ($B = 2.97$, $p = 0.009$), outpatient department ($B = 3.93$, $p = 0.021$), inpatient department ($B = 2.50$, $p = 0.001$), previous VTE prevention training ($B = 3.46$, $p < 0.001$), VTE prevention management in hospital ($B = 4.93$, $p < 0.001$), better knowledge ($B = 0.04$, $p = 0.038$), and positive attitude towards VTE prevention ($B = 1.35$, $p < 0.001$) were all significantly and positively associated with higher practice scores in VTE prevention.

**Conclusions**. Our study provided a comprehensive understanding of the ophthalmic nurses' knowledge, attitude, and practice in VTE prevention, as well as identified specific items in each dimension for improvement. In addition, our study showed multiple factors were associated with ophthalmic nurses' practice in VTE prevention, including environmental factors, training and management, knowledge and attitudes toward VTE prevention. Our findings provide important implications and guidance for future intervention programs to improve the ophthalmic nurses' knowledge, attitude, and practice in VTE prevention.

## INTRODUCTION

Venous thromboembolism (VTE), a disease entity comprising deep vein thrombosis (DVT) and pulmonary embolism (PE), is a leading cause of morbidity and mortality in hospitalized patients (*Kahn et al., 2018*; *Tritschler et al., 2018*; *Anderson et al., 2019*). Globally, VTE causes a significant disease burden, affecting nearly 10 million people a year (*ISTH Steering Committee for World Thrombosis Day, 2014*; *Khan et al., 2021*). The estimated annual healthcare costs for VTE ranged between €1.5–3.3 billion in Europe, and US $7–10 billion in the USA (*Barco et al., 2016*; *Grosse et al., 2016*). Hence, VTE prevention ranked top one of the 79 patient safety practice strategies in hospitals (*Shojania et al., 2001*).

Although the incidence of VTE is relatively low in Asian counties, including China, it has increased steadily in recent years (*Liew et al., 2017*). Between 2007 and 2016, hospitalization rates for VTE increased from 3.2 to 17.5 per 100,000 people in China (*Zhang et al., 2019*). Several reasons may account for the apparent rise, including population aging, lifestyle changes, increasing awareness, and better detection (*Liew et al., 2017*; *Zhang et al., 2019*; *Di Nisio, Van & Buller, 2016*). Therefore, over the last decade, China has developed a variety of evidence-based consensus, guidelines, and national programs to focus on VTE prevention and treatment (*Expert Committee of the National Capacity Building Project for Prevention and Treatment of Pulmonary Embolism and Deep Vein Thrombosis, 2022*; *Li, Zhang & Wang, 2017*; *The National Health Commission , 2021*).

Although VTE among ophthalmic surgical patients is less reported, a growing body of evidence suggests that ophthalmic surgical patients are at an increased risk of VTE (*Kaplan & Reba, 1972*; *Huang & Wei, 2016*; *Brill et al., 2019*; *Law, Chan & Cheng, 2018*; *Wang et al., 2016*). Studies have shown that VTE is the leading cause of death in patients undergoing ophthalmic surgery (*Kaplan & Reba, 1972*; *Kanbay et al., 2011*). Most ophthalmic surgical patients are middle-aged or older adults (*Lei et al., 2019*) and are comorbid with other chronic illnesses such as obesity, coronary heart disease, and diabetes (*Liu et al., 2017*; *Gong et al., 2021*), which are all well-known risk factors for VTE. In addition, most ophthalmic surgeries take more than 30 min and the prolonged immobility in a postoperative prone

position after retina surgery also predisposes patients to VTE (*Brill et al., 2019*). Therefore, VTE prevention for ophthalmic surgical patients deserves closer attention.

Most VTE is preventable (*Khan et al., 2021*; *Liew et al., 2017*; *Cohen et al., 2007*). It is safer and more cost-effective to prevent the occurrence of VTE than to treat VTE after it develops (*Khan et al., 2021*). However, the implementation of VTE prevention remains inadequate or inappropriate (*Kahn et al., 2018*). As the front-line health workers providing the most direct care and spending the most time with patients, nurses play a crucial role in the effective identification, prevention, and management of VTE in clinical practice. Hence, it is important for nurses to clearly understand the therapeutic and prophylactic measures of thrombosis in order to adequately prevent VTE in hospitalized patients. A number of studies have demonstrated that correct knowledge, a positive attitude, and effective practice on VTE prevention are vital for the prevention and management of VTE among patients in various conditions (*Wang et al., 2021*; *Oh, Boo & Lee, 2017*; *Ma et al., 2018*). However, to our knowledge, no study has ever explored the KAP of VTE prevention among ophthalmic nurses, despite the increased risk of VTE associated with ophthalmic surgery.

To bridge the research gap, we conducted the current study to (a) examine the KAP of ophthalmic nurses towards VTE prevention, and (b) evaluate factors associated with the practice of ophthalmic nurses regarding VTE prevention. Our findings would provide a scientific basis for the development of future KAP training programs targeted at VTE prevention among ophthalmic nurses to prevent VTE in ophthalmic surgery patients.

## MATERIALS AND METHODS

### Design and settings

This cross-sectional study was conducted from March 10 to April 30 2021 among ophthalmic nurses in Hunan and Zhejiang Provinces, China. Portions of this text were previously published as part of a preprint/thesis (*Zhou et al., 2023*).

Hunan Province is located in south-central China and includes 13 prefecture-level cities and one autonomous prefecture. Its medical resources in ophthalmology rank at a middle level in China (*Jing et al., 2019*), serving a population of more than 66 million registered residents (http://tjj.hunan.gov.cn/hntj/m/jmxx_1/202204/t20220422_22744136.html). Zhejiang Province is located in the southeast of China and includes 11 prefecture-level cities. Its medical resources in ophthalmology rank at a high level in China (*Jing et al., 2019*), serving a population of more than 65 million registered residents (http://tjj.zj.gov.cn/art/2022/2/22/art_1229129205_4882102.html).

### Participants

The sample size was calculated according to the form for cross-sectional study:

$n = Z_{2\alpha/2}P(1-P)/\delta^2$, where $P$ (the correct rate of VTE prevention knowledge) was estimated at 73.4% based on a previous study (*Wang et al., 2021*), Z was set as 1.96 at a confidence interval of 95%, allowable error $\delta$ was set as 4%. Considering a rejection or loss-to-follow-up rate of 20%, we expanded our final sample size to 563. After screening

the nurses for inclusion and exclusion criteria, a total of 647 nurses were enrolled in the study.

Inclusion criteria for participants included: (a) aged ≥18 years old; (b) obtaining a Nursing Practice Certificate issued by the Ministry of Health of China; (c) currently working in ophthalmology for more than one year; (d) with normal mental capacity and were able to complete the questionnaire. Participants were excluded if they were: (a) part-time employees; (b) in-training nurses.

## Procedure and ethics approval

The questionnaire was distributed online using an electronic link (Sojump online platform, https://www.wjx.cn/) through the official channels of the Ophthalmology Nursing Committee of Hunan and Zhejiang Medical Association. The questionnaire link included a cover letter explaining the purpose of the study as well as the informed consent form. The questionnaire was designed in such a way that it could not be submitted until all questions were answered. Participants could review and change their responses before submission.

The Medical Ethics Committee of the Xiangya Hospital, Central South University approved this study protocol before the formal survey (approval number: 202103044). Participation in this survey was voluntary and anonymous. Participants had to answer a yes-no question to confirm their willingness to participate voluntarily. After providing electronic informed consent, the participant was directed to complete the self-reported questionnaire. All the data was kept confidential and processed anonymously. The electronic survey platform was made available for about two months. During this time, a total of 647 ophthalmic nurses participated in this survey, and 610 nurses responded effectively, yielding a response rate of 94.28%.

## Measurements

The questionnaire was developed based on a literature review, the latest guidelines regarding care for VTE, and expert advice. The newly developed questionnaire was evaluated and reviewed with respect to methodology and content by a focus group of experts in VTE prevention ($n = 2$), medicine ($n = 2$), biostatistics ($n = 2$), and nursing management ($n = 2$). The questionnaire showed good content validity, with an index of 0.894. Before the formal investigation, a pilot study was conducted on 26 nurses to assess internal consistency and test-retest reliability (after two weeks). The Cronbach's alpha coefficient and test-retest reliability of the questionnaire were 0.875 and 0.912, respectively, indicating good reliability. The questionnaire took approximately 10–20 min to complete.

The final questionnaire contains 58 items in two sections: the socio-demographic (14 items) and KAP (44 items) sections. The socio-demographic section includes questions on age, gender, province, city, hospital level, type of hospital, service years, professional title, education, position, department, VTE prevention management in hospital, previous VTE training, and source of VTE knowledge. The KAP section includes three subscales on VTE prevention: knowledge (27 items), attitude (five items), and practice (12 items). The knowledge subscale includes 27 single-choice questions (true, false, and do not know), with 5 points assigned to each correct answer and 0 points assigned to each incorrect

or unknown answer. The 27 items reflect six themes: definition, pathogeny, risk factors, clinical manifestations, assessment tools, and prevention strategies for VTE. Sample items include "Post-thrombotic syndrome is the most severe complication of DVT" and "The Caprini risk assessment model of high VTE risk partition is 3∼4 points". Among them, ten items are reverse-worded questions. The total knowledge score ranges from 0 to 135, with a higher score denoting better knowledge of VTE prevention. The attitude scale includes five items scored on a 5-point Likert scale, ranging from 1 (strongly disagree) to 5 (strongly agree). The five items reflect four themes: responsibility for VTE prevention, necessity for VTE prevention, attitude toward taking part in VTE training, and significance of VTE prevention. Sample items include "It is necessary to train nurses in VTE prevention". The total attitude score ranges from 5 to 25, with a higher score indicating a more positive attitude toward VTE prevention. The practice subscale includes 12 items scored on a 5-point Likert scale ranging from 1 (never) to 5 (always). The 12 items reflect four themes: assessment, taking measures for VTE prevention, giving feedback on the occurrence of VTE to a doctor, and medication nursing. Sample items include "Assessing the lower limbs of perioperative patients regularly in nursing practice". The total practice score ranges from 12 to 60, with a higher score indicating more active practice in VTE prevention.

### Data analysis

SPSS IBM version 25.0 (IBM Corp., Chicago, IL, USA) was used to perform all analyses. Descriptive statistics were expressed as numbers and percentages or as means ± SDs (standard deviations).

The independent $t$-test, one-way ANOVA, or Kruskal-Wallis H test were performed to examine differences in KAP scores by demographic information, as appropriate. Scheffe's test was adopted as a post-test of one-way ANOVA. Pearson's correlation coefficients were computed to examine the relationships between knowledge, attitude, and practice toward VTE prevention. Additionally, the multiple linear regression model with the forward method was used to analyze the influencing factors of ophthalmic nurses' practice in VTE prevention. Categorical variables were transformed into dummy variables. The statistical significance level was set at $p < .05$ (two-sided).

## RESULTS

### KAP scores by sample characteristics

A total of 610 ophthalmic nurses completed the questionnaire. They were distributed in 17 prefecture-level cities in Hunan and Zhejiang provinces. The mean age of the participants was 33.02 ± 6.83 and ranged from 20 to 57 years. Table 1 shows the socio-demographic characteristics of participants and comparisons of KAP scores by sample characteristics. Most were female (99.3%), had a bachelor's degree (83.6%), worked in tertiary-level hospitals (89%), and had received VTE prevention training (71%). However, 23.9% of participants reported that their hospital did not carry out VTE prevention and control management. In addition, comparisons of knowledge, attitude, and practice scores by sample characteristics consistently showed significant differences in the following variables:

hospital level, type of hospital, department, previous VTE training, VTE prevention management in hospital, and previous VTE nursing experience.

## Descriptive of KAP scores

Table 2 shows the three items with the lowest scores in each dimension of the knowledge, attitude, and practice assessment scale. The mean knowledge score was $104.45 \pm 19.97$ out of a maximum score of 135, with a correct rate of 77.4%. The three lowest-scored knowledge items were related to VTE complications ($1.14 \pm 2.10$), indications for physical prevention ($1.39 \pm 2.24$), and risk assessment ($2.24 \pm 2.49$). In addition, the most reported source of knowledge on VTE prevention was from their hospital or department training (83.8%), followed by the Internet (46.1%) and guideline or literature reading (42.6%), and about 64% of the respondents reported more than one source. The mean attitude score was $21.96 \pm 2.72$ out of a maximum score of 25. The three lowest-scored attitude items were related to the necessity of training nurses on VTE prevention ($4.32 \pm 0.79$), VTE risk ($4.36 \pm 0.89$), and the necessity of educating patients and families on VTE prevention ($4.41 \pm 0.65$). Regarding practice for VTE prevention, the mean score was $48.96 \pm 11.23$ out of a maximum score of 60, with a correct rate of 81.6%. The three lowest scored items were related to the use of the VTE risk assessment scale ($3.50 \pm 1.43$), assessment of VTE ($3.83 \pm 1.25$), and education of patients and families about VTE prevention ($4.04 \pm 1.24$).

## Correlations between knowledge, attitude, and practice

Pearson's correlation results listed in Table 3 revealed that the knowledge, attitude, and practice scores of ophthalmic nurses regarding VTE prevention correlated significantly with each other. The knowledge score was positively and weakly associated with attitude ($r = 0.244$, $p < 0.01$) and practice ($r = 0.293$, $p < 0.001$). Furthermore, the attitude score had a moderate positive association with practice ($r = 0.413$, $p < 0.01$).

## Influencing factors of VTE prevention practice

Table 4 shows the results of the multiple linear regression on the influencing factors of ophthalmic nurses' practice in VTE prevention, with all significant socio-demographic variables in univariate analysis, knowledge, and attitude as independent variables. Among the 10 variables included in the model, seven remained significant after controlling for all other confounders: province, hospital type, department, previous VTE prevention training, VTE prevention management in the hospital, knowledge, and attitude towards VTE prevention. Practice scores were significantly higher in Hunan Province than Zhejiang Province ($B = 2.77$, $p = 0.006$), in general hospitals than specialty hospitals ($B = 2.97$, $p = 0.009$), in outpatient departments than operating rooms ($B = 3.83$, $p = 0.021$), and in inpatient departments than operating rooms ($B = 2.50$, $p = 0.001$). In addition, previous VTE prevention training ($B = 3.46$, $p < 0.001$), VTE prevention management in hospital ($B = 4.93$, $p < 0.001$), better knowledge towards VTE prevention ($B = 0.04$, $p < 0.038$), and a positive attitude towards VTE prevention ($B = 1.35$, $p < 0.001$) were all significantly and positively associated with higher practice scores in VTE prevention.

**Table 1 Socio-demographic characteristics and correlations with knowledge, attitude, and practice regarding VTE prevention.**

| Variables | Frequency (*n*, %) | Knowledge Mean ± SD | Test results | Attitude Mean ± SD | Test results | Practice Mean ± SD | Test results |
|---|---|---|---|---|---|---|---|
| **Age (years)** | | | | | | | |
| ≤29 | 189 (31.0) | 106.03 ± 17.64 | $H = 9.373$ | 22.05 ± 2.81 | $H = 3.683$ | 50.30 ± 10.74 | $H = 4.439$ |
| 30~39 | 319 (52.3) | 102.12 ± 22.08 | $p = 0.009^*$ | 22.03 ± 2.74 | $p = 0.159$ | 48.02 ± 11.76 | $p = 0.109$ |
| ≥40 | 102 (16.7) | 108.77 ± 15.82 | | 21.59 ± 2.48 | | 49.40 ± 10.16 | |
| **Gender** | | | | | | | |
| Male | 4 (0.7) | 113.75 ± 22.87 | $t = .934$ | 23.50 ± 1.29 | $t = 1.135$ | 56.25 ± 3.50 | $t = 1.305$ |
| Female | 606 (99.3) | 104.39 ± 19.96 | $p = 0.350$ | 21.95 ± 2.73 | $p = 0.257$ | 48.91 ± 11.25 | $p = 0.193$ |
| **Province** | | | | | | | |
| Zhejiang | 281 (46.1) | 102.88 ± 20.89 | $t = -1.796$ | 21.78 ± 2.64 | $t = -1.567$ | 47.85 ± 11.86 | $t = -2.255$ |
| Hunan | 329 (53.9) | 105.79 ± 19.08 | $p = 0.073$ | 22.12 ± 2.78 | $p = 0.118$ | 49.90 ± 10.58 | $p = 0.025^*$ |
| **Hospital level** | | | | | | | |
| First-level hospital | 7 (1.2) | 63.57 ± 30.92 | $H = 15.481$ | 17.86 ± 5.82 | $H = 7.460$ | 35.43 ± 14.13 | $H = 17.443$ |
| Secondary hospital | 61 (10.0) | 100.57 ± 22.18 | $p < 0.05^*$ | 21.69 ± 2.94 | $p = 0.024^*$ | 44.92 ± 12.17 | $p < 0.05^*$ |
| Tertiary hospital | 542 (88.8) | 105.42 ± 18.96 | | 22.05 ± 2.60 | | 49.58 ± 10.89 | |
| **Hospital type** | | | | | | | |
| Specialist hospital | 193 (31.6) | 99.25 ± 22.01 | $t = -4.444$ | 21.47 ± 2.94 | $t = -3.052$ | 44.53 ± 13.20 | $t = -6.873$ |
| General hospital | 417 (68.4) | 106.86 ± 18.49 | $p < 0.05^*$ | 22.19 ± 2.59 | $p = 0.002^*$ | 51.00 ± 9.53 | $p < 0.05^*$ |
| **Service years** | | | | | | | |
| ≤5 | 137 (22.5) | 107.19 ± 16.90 | $H = 17.231$ | 21.96 ± 2.49 | $H = 4.576$ | 49.85 ± 10.59 | $H = 3.131$ |
| 6~10 | 182 (29.8) | 104.73 ± 19.59 | $p = 0.001^*$ | 22.25 ± 2.51 | $p = 0.206$ | 49.45 ± 11.45 | $p = 0.372$ |
| 11~15 | 154 (25.2) | 98.83 ± 23.85 | | 21.99 ± 2.88 | | 47.80 ± 12.00 | |
| ≥16 | 137 (22.5) | 107.66 ± 17.18 | | 21.54 ± 2.98 | | 48.71 ± 10.63 | |
| **Professional title** | | | | | | | |
| Primary | 313 (51.3) | 102.96 ± 20.96 | $H = 14.707$ | 21.89 ± 2.58 | $H = 2.101$ | 49.22 ± 11.42 | $H = 1.387$ |
| Intermediate | 251 (41.2) | 104.86 ± 18.82 | $p = 0.001^*$ | 22.12 ± 2.75 | $p = 0.350$ | 48.78 ± 11.11 | $p = 0.500$ |
| Senior | 46 (7.5) | 112.39 ± 17.38 | | 21.54 ± 3.42 | | 48.15 ± 10.67 | |
| **Highest education attained** | | | | | | | |
| Associate degree and below | 75 (12.3) | 98.67 ± 24.46 | $H = 14.630$ | 21.25 ± 2.68 | $H = 9.665$ | 48.96 ± 12.40 | $H = 0.607$ |
| Bachelor's degree | 510 (83.6) | 104.83 ± 18.95 | $p = 0.001^*$ | 22.05 ± 2.62 | $p = 0.008^*$ | 49.10 ± 10.85 | $p = 0.738$ |
| Master's degree and above | 25 (4.1) | 114.00 ± 21.36 | | 22.24 ± 4.28 | | 46.04 ± 14.75 | |
| **Position** | | | | | | | |
| Nurse | 543 (89.0) | 103.68 ± 20.10 | $t = -2.717$ | 22.02 ± 2.66 | $t = 1.595$ | 49.27 ± 11.14 | $t = 2.001$ |
| Head nurse | 67 (11.0) | 110.67 ± 17.84 | $p = 0.007^*$ | 21.46 ± 3.15 | $p = 0.111$ | 46.37 ± 11.64 | $p = 0.046^*$ |
| **Department** | | | | | | | |
| Operating room | 52 (8.5) | 98.65 ± 19.63 | $F = 18.753$ | 21.44 ± 3.44 | $F = 5.528$ | 40.35 ± 14.22 | $F = 25.036$ |
| Outpatient department | 83 (13.6) | 93.80 ± 26.42 | $p < 0.05^*$ | 21.19 ± 2.80 | $p = 0.004^*$ | 45.66 ± 13.20 | $p < 0.05^*$ |
| Inpatient department | 475 (77.9) | 104.45 ± 19.97 | | 22.15 ± 2.59 | | 50.47 ± 9.88 | |

**Table 1** (*continued*)

| Variables | Frequency (*n*, %) | Knowledge | | Attitude | | Practice | |
|---|---|---|---|---|---|---|---|
| | | Mean ± SD | Test results | Mean ± SD | Test results | Mean ± SD | Test results |
| Previous VTE prevention training | | | | | | | |
| No | 176 (28.8) | 93.21 ± 24.82 | $t = -9.477$ | 21.58 ± 3.09 | $t = -2.220$ | 43.22 ± 12.75 | $t = -8.489$ |
| Yes | 434 (71.2) | 109.01 ± 15.49 | $p < 0.05^*$ | 22.12 ± 2.54 | $p = 0.027^*$ | 51.28 ± 9.63 | $p < 0.05^*$ |
| VTE prevention management in hospital | | | | | | | |
| No | 146 (23.9) | 94.38 ± 26.59 | $t = -7.276$ | 21.37 ± 3.17 | $t = -3.038$ | 41.50 ± 13.60 | $t = -9.908$ |
| Yes | 464 (76.1) | 107.62 ± 16.16 | $p < 0.05^*$ | 22.15 ± 3.79 | $p < 0.05^*$ | 51.30 ± 9.21 | $p < 0.05^*$ |
| Previous VTE nursing experience | | | | | | | |
| No | 487 (79.8) | 103.21 ± 20.78 | $t = -3.066$ | 21.77 ± 2.76 | $t = -3.506$ | 47.76 ± 11.69 | $t = -5.353$ |
| Yes | 123 (20.2) | 109.35 ± 15.47 | $p = 0.002^*$ | 22.72 ± 2.42 | $p < 0.05^*$ | 53.69 ± 7.53 | $p < 0.05^*$ |

**Notes.**
*The independent *t*-test, one-way ANOVA, and Kruskal-Wallis *H* test $p < 0.05$.
SD, Standard Deviations; VTE, venous thromboembolism.

**Table 2** The three items with the lowest scores in each dimension of the knowledge, attitude, and practice assessment scale for VTE prevention among ophthalmic nurses.

| Dimension | Items | Scores | Correct rate (%) |
|---|---|---|---|
| Knowledge | K4. Post-thrombotic syndrome is the most severe complication of DVT. | 1.14 ± 2.10 | 22.8 |
| | K14. Physical prevention of VTE can be used for patients with congestive heart failure and severe edema of lower limbs. | 1.38 ± 2.24 | 27.5 |
| | K8. The Caprini risk assessment model of high VTE risk partition is 3 ∼4 points. | 2.24 ± 2.49 | 44.8 |
| attitude | A5. It is necessary to train nurses in VTE prevention. | 4.32 ± 0.79 | - |
| | A1. VTE will increase nursing risk and even cause medical disputes. | 4.36 ± 0.89 | - |
| | A3. Nurses should carry out health education on VTE prevention for patients and their families. | 4.41 ± 0.65 | - |
| Practice | P1. Using the VTE risk assessment scale properly. | 3.50 ± 1.43 | 70.0 |
| | P8. Assessing the lower limbs of perioperative patients regularly in nursing practice. | 3.83 ± 1.25 | 76.6 |
| | P6. Carrying on health education about VTE prevention for patients and their families. | 4.04 ± 1.24 | 80.8 |

**Notes.**
DVT, Deep Vein Thrombosis; VTE, Venous Thromboembolism.

**Table 3** Correlation matrix for knowledge, attitude, and practice regarding VTE prevention of ophthalmic nurses.

| Variables | VTE prevention knowledge | Attitude towards VTE prevention | VTE prevention practice |
|---|---|---|---|
| VTE prevention knowledge | 1 | | |
| attitude toward VTE prevention | $0.244^*$ | 1 | |
| VTE prevention practice | $0.293^*$ | $0.413^*$ | 1 |

**Notes.**
*$p < 0.05$.

**Table 4** **Multiple linear regression analysis of demographic characteristics, knowledge, and attitude associated with the practice of VTE prevention.**

| Independent variables | B coefficient | Standard coefficient | B-value | Statistics | P-value |
|---|---|---|---|---|---|
| Province | 2.772 | 1.006 | 0.123 | 2.755 | 0.006[*] |
| Hospital level | −0.216 | 1.140 | −0.007 | −0.190 | 0.850 |
| Hospital type | 2.965 | 1.135 | 0.123 | 2.613 | 0.009[*] |
| Position | −2.278 | 1.217 | −.064 | −1.872 | 0.062 |
| Department | | | | | |
|    Operating room | – | – | – | – | – |
|    Outpatient department | 3.829 | 1.656 | 0.117 | 2.313 | 0.021[*] |
|    Inpatient department | 2.497 | 0.722 | 0.185 | 3.459 | 0.001[*] |
| Previous VTE prevention training | 3.460 | 0.957 | 0.140 | 3.615 | <0.05[*] |
| VTE prevention management in hospital | 4.929 | 1.084 | 0.187 | 4.547 | <0.05[*] |
| Previous VTE nursing experience | 1.762 | 0.983 | 0.063 | 1.793 | 0.073 |
| Knowledge of VTE prevention | 0.044 | 0.021 | 0.079 | 2.084 | 0.038[*] |
| attitude toward VTE prevention | 1.351 | 0.143 | 0.327 | 9.444 | <0.05[*] |

**Notes.**
[*]$p < 0.05$.
VTE, venous thromboembolism.

## DISCUSSION

The current study investigated ophthalmic nurses' knowledge, attitude, and practice levels toward venous thromboembolic prevention and examined the factors associated with ophthalmic nurses' practice. Nurses' KAP is critical for VTE prevention and control and is paramount for intervention development and policy implementation. To our knowledge, this is the first KAP survey of venous thromboembolic prevention among ophthalmic nurses. The results showed that the ophthalmic nurses had a relatively higher level of KAP in VTE prevention compared to previous studies on nurses from other specialties (*Wang et al., 2021*; *Oh, Boo & Lee, 2017*; *Ma et al., 2018*; *Yan et al., 2021*; *Xu et al., 2017*). However, nurses had low scores in certain items of each dimension, indicating the need for improvement. In addition, province, hospital type, department, previous VTE prevention training, VTE prevention management in hospitals, and attitude towards VTE prevention were significantly associated with ophthalmic nurses' practice.

Our study showed that the ophthalmic nurses' KAP towards VTE prevention was at a desirable level, which may be explained by two reasons (*Ay, Pabinger & Cohen, 2017*). First, the Chinese government's involvement has played a significant role in the promotion and implementation of VTE prevention. VTE prevention was listed among the top ten aims of the National Medical Quality and Safety Improvement Targets for 2021 (*The National Health Commission , 2021*), which provided significant momentum for developing VTE prevention. Secondly, the high score of the overall KAP might be due to the respondent's characteristics. In our study, most respondents came from advanced hospitals with more training opportunities and better management systems for VTE prevention.

Although the ophthalmic nurses showed a satisfactory correct rate in VTE prevention knowledge (76.4%), they held some misunderstandings in the following three items with

low correct rates: DVT complication (22.8%), the physical prevention of VTE (27.5%), and the Caprini risk assessment model (44.8%). These findings suggest that there is still room for improvement in these areas related to VTE prevention, which may be trained through professional courses. Our findings indicated that attending professional training in a hospital or department is one of the main sources of knowledge for participants (83.8%). Furthermore, Lee and colleagues have reported that nurses with previous VTE training had higher self-efficacy in VTE assessment and prevention (*Lee et al., 2014*). Therefore, it is suggested that providing high-quality training in VTE prevention may effectively improve ophthalmic nurses' knowledge of VTE prevention. The training can take multiple forms, such as academic workshops, knowledge lectures, and nursing rounds, which may not only strengthen the employees' professional knowledge but also the management system (*Bartholomew, 2017*; *Ortel et al., 2020*).

Our study showed that the ophthalmic nurses' attitudes toward VTE prevention were at a desirable level. The nurses' positive attitude toward VTE prevention may be explained by their high level of VTE prevention knowledge, as studies have consistently shown a positive association between knowledge and attitude (*Zhang et al., 2021*). As for practice in VTE prevention, our study showed a favorable correct rate of 81.6%, indicating most nurses have been taking active actions to prevent VTE. However, they had low performance in the following two areas: prevention assessment of risks and health education for the high-risk population, indicating room for improvement in these areas. Further multivariate regression analysis identified seven influencing factors that were associated with nurses' practice in VTE prevention, which may be classified into the following three categories: (a) environmental factors such as province, hospital type, and department; (b) training and management; and (c) knowledge and attitude toward VTE prevention, each discussed below:

In terms of environmental factors, the practice scores in VTE prevention were significantly higher in Hunan Province than in Zhejiang Province, in general hospitals than in specialty hospitals, and in inpatient or outpatient departments than in operating rooms. These differences may reflect the unequal distribution of prevention resources, training systems, and management systems. Furthermore, it may also be caused by the unequal distribution of high-risk VTE patients. For example, general hospitals are more likely to admit patients with comorbidities or complications that predispose them to VTE risk. As a result, nurses in general hospitals encounter VTE patients more frequently and thus have higher practice in VTE.

In terms of training and management, nurses who received previous VTE prevention training and had VTE prevention management in the hospital had significantly higher practice scores than those without. These findings were consistent with previous studies showing the effectiveness of strengthening training and management in improving nurses' KAP towards VTE prevention (*Yan et al., 2021*; *Zhang et al., 2021*). These findings suggest that it is essential to provide regular and professional academic courses or conferences on VTE prevention for nurses, which not only requires support and arrangement from the hospital but also a better management system. In order to better evaluate the effect of training and make appropriate adjustments, there need to be measurable goals for each

activity and timely feedback from nurses (*Maynard & Stein, 2010*). From the hospital's perspective, the systematic redesign of the VTE training system is crucial to VTE prevention and improvement. In addition, the managers' attention could be a facilitator for the organizational climate (*Bartholomew, 2017*; *Jönsson et al., 2018*), which is conducive to the improvement of nurses' KAP level for VTE prevention. Therefore, the managers' involvement is key to VTE management.

In terms of knowledge and attitude towards VTE prevention, nurses with better knowledge and a positive attitude toward VTE prevention had significantly higher scores in VTE prevention practice, a finding also consistent with previous studies (*Yan et al., 2021*; *Zhang et al., 2021*). A positive attitude indicates high levels of self-efficacy and responsibility regarding VTE prevention. Nurses with positive attitudes toward VTE prevention had higher awareness of the importance of self-education as well as patient education on VTE prevention (*Lee et al., 2014*). However, according to the present data, nearly 20% of nurses did not provide timely and necessary health education for patients, which may lead to an increased risk of VTE and related morbidity and mortality (*Bauer et al., 2019*; *Lavall & Costello, 2015*). Our findings suggest that more education and support may be needed to improve nurses' knowledge and attitudes toward VTE prevention, especially in the aspect of patient education. In addition, the management system should be supportive and effective in optimizing the nurse workflow, which can start with improving the nurse-patient ratio and reducing unnecessary paperwork to allow more bedside time for the nurse. A better environment and lower workload may help foster better knowledge and a positive attitude among nurses, which will consequently lead to improved practice in VTE prevention (*Van Bogaert et al., 2013*).

## LIMITATIONS

This study has several potential limitations. First, the nurses were mostly recruited from tertiary hospitals in two provinces using non-random sampling methods, which may limit the sample representativeness and the result generalizability. Second, the cross-sectional study design may preclude any causal inferences between nurses' practice in VTE prevention and its influencing factors. Third, the assessment of KAP was based on a newly developed measurement tool instead of an internationally standard tool, making it difficult to make comparisons with other studies. Fourth, all data collection relied on nurses' self-report, which may be subject to recall bias and social desirability bias. Finally, although the knowledge, attitudes, and practice subscale scores showed significant associations with each other, all the correlation coefficients were below 0.60, which may be related to the large sample size and thus warrants further examination in future studies.

### Relevance to clinical practice

Ophthalmic nurses have insufficient knowledge of complications, risk assessment, and indications about VTE prevention, low confidence in risk assessment and health education for the high-risk population, and inadequate practice in risk assessment and education regarding VTE prevention. The key solution for the above issues is to promote ongoing education and training based on organization and nursing leadership. The findings of the

study also highlight that having a well-trained nurse manage and prevent VTE is vital to ensuring better patient outcomes.

## CONCLUSIONS

Our study provided a comprehensive understanding of the ophthalmic nurses' knowledge, attitude, and practice in VTE prevention, as well as identified specific items in each dimension for improvement. In addition, our study showed multiple factors were associated with ophthalmic nurses' practice in VTE prevention, including environmental factors, training and management, knowledge and attitudes toward VTE prevention. Our findings provide important implications and guidance for future intervention programs to improve the ophthalmic nurses' knowledge, attitude, and practice in VTE prevention.

## ACKNOWLEDGEMENTS

We would like to acknowledge all the persons and participants involved in the study.

### Funding
The authors received no funding for this work.

### Competing Interests
The authors declare there are no competing interests.

### Author Contributions

- Xiaoxi Zhou conceived and designed the experiments, performed the experiments, authored or reviewed drafts of the article, and approved the final draft.
- Minhui Dai conceived and designed the experiments, performed the experiments, authored or reviewed drafts of the article, and approved the final draft.
- Lingyu Sun performed the experiments, authored or reviewed drafts of the article, and approved the final draft.
- Chunyan Li conceived and designed the experiments, performed the experiments, authored or reviewed drafts of the article, and approved the final draft.
- Wendi Xiang performed the experiments, authored or reviewed drafts of the article, and approved the final draft.
- Yaoyao Lin analyzed the data, prepared figures and/or tables, and approved the final draft.
- Dandan Jiang analyzed the data, prepared figures and/or tables, and approved the final draft.

### Ethics

The following information was supplied relating to ethical approvals (*i.e.*, approving body and any reference numbers):

The study involving human participants was reviewed and approved by The Medical Ethics Committee of the Xiangya Hospital, Central South University, Changsha, China (approval number: 202103044).

## Data Availability

The raw measurements are available in the Supplemental Files.

## Supplemental Information

Supplemental information for this article can be found online at http://dx.doi.org/10.7717/peerj.15947#supplemental-information.

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
