# Peer review of "Ophthalmic nurses' knowledge, attitude, and practice toward venous thromboembolic prevention: a dual-center cross-sectional survey"

_PeerJ, doi:10.7717/peerj.15947_

## Round 0.1 · original submission · Minor Revisions

The whole language is needed to be revised by a native English speaker.

Reviewer 1 ·

Basic reporting

The work is not enough for the conclusion.

Experimental design

The work is not enough for the conclusion.

Validity of the findings

The work is not enough for the conclusion.

Additional comments

The work is not enough for the conclusion.

Reviewer 2 ·

Basic reporting

I would like to thank the authors for the research done. Overall the manuscript is well-written and provides a valuable context to the literature. The language is clear and the overall structure is professional.

The authors did a great job describing their rationale for the study and sharing detailed insight into their methods and results.

I have noticed some minor edits in terms of language and formatting I added notes to the PDF referencing them, otherwise, the researchers did a great job reporting their study.

Experimental design

The study is well-designed and the researchers did a great job sharing possible limitations.

I just noticed that the researchers did state that they used a convenience sample, early in the manuscript, when later under the sample size, they estimated the sample size.

This point is confusing, the researchers need to be clear on their sample size whether it is a convenience sample or calculated, and what method they used for their calculation.

Also, it would be great if you would add more clarification on the Knowledge question's wording (were there any reverse-worded questions?), the same with the attitude and practice questions

Validity of the findings

This study brings attention to a leading cause of preventable mortality related to surgical intervention. Having a well-trained nurse to manage and prevent the condition is vital to ensure better patient outcomes. Again, I would like to thank the researchers for their work and what can it bring to the prevention practices i

The tables need to have a footnote with the statistical test used
The raw data, practice questions 11 and 12 look like they are the same. I would appreciate clarification on this.

Additional comments

I have no more comments

Annotated reviews are not available for download in order to protect the identity of reviewers who chose to remain anonymous.

Reviewer 3 ·

Basic reporting

1. It is a large scale well performed study.

2. In some places it is difficult to understand what the authors mean which I have highlighted and also there is some language errors which need to be checked before submitting next version.

3. The article is well structured with clear from Abstract to conclusion.

Experimental design

1. Even though this study used a convenience sample, it has well planned methods.

2. Research question is well defined and important for this field.

3. It is easy to read the methods and understand.

Validity of the findings

1. The statistical software and analyses performed is sound.

2. Sometimes there is too many p values, instead one suggestion is to use <0.05 when it is significant and >0.05 when it is not then providing exact numbers.

3. Even though there was a statistical significant correlation (which probably is due to large sample) interpretation of correlation coefficient <0.6 must be used with caution.

Additional comments

Please see attached pdf where I have highlighted some texts and also added my comments with text box.

Annotated reviews are not available for download in order to protect the identity of reviewers who chose to remain anonymous.

---

## Round 0.2 · accepted · Accept

The authors' reply meets the standard.

Reviewer 2 ·

Basic reporting

The researchers addressed all comments. Thanks!

Experimental design

The researchers addressed all comments. Thanks!

Validity of the findings

The researchers addressed all comments. Thanks!

Reviewer 3 ·

Basic reporting

All my previous comments and suggestions are answered.

Thanks.

Experimental design

All my previous comments and suggestions are answered.

Thanks.

Validity of the findings

All my previous comments and suggestions are answered.

Thanks.

Additional comments

All my previous comments and suggestions are answered.

Thanks.